# Antioxidant and Anti-Inflammatory Properties of Conceivable Compounds from *Glehnia littoralis* Leaf Extract on RAW264.7 Cells

**DOI:** 10.3390/nu16213656

**Published:** 2024-10-27

**Authors:** Min Yeong Park, Hun Hwan Kim, Se Hyo Jeong, Pritam Bhangwan Bhosale, Abuyaseer Abusaliya, Hyun Wook Kim, Je Kyung Seong, Kwang Il Park, Gon Sup Kim

**Affiliations:** 1Research Institute of Life Science, College of Veterinary Medicine, Gyeongsang National University, Gazwa, Jinju 52828, Republic of Korea; lilie17@daum.net (M.Y.P.); shark159753@naver.com (H.H.K.); tpgy123@gmail.com (S.H.J.); shelake.pritam@gmail.com (P.B.B.); yaseerbiotech21@gmail.com (A.A.); kipark@gnu.ac.kr (K.I.P.); 2Division of Animal Bioscience & Integrated Biotechnology, Jinju 52725, Republic of Korea; hwkim@gnu.ac.kr; 3Laboratory of Developmental Biology and Genomics, BK21 PLUS Program for Creative Veterinary Science Research, Research Institute for Veterinary Science, College of Veterinary Medicine, Seoul National University, Seoul 08826, Republic of Korea; snumouse@snu.ac.kr

**Keywords:** COX-2, iNOS, MAPK, NF-κB, antioxidant, *Glehnia littoralis*, HPLC-MS/MS

## Abstract

Background/Objectives: *Glehnia littoralis* is a medicinal plant, but the scientific basis is still unclear. This study thoroughly investigated phenols from *Glehnia littoralis* extract (GLE) to determine their potential as anti-inflammatory and antioxidant agents. Methods: High-performance liquid chromatography (HPLC) and mass spectrometry (MS) were used to analyze the compounds in GLE. In addition, we performed GLE in vitro in macrophages after lipopolysaccharide (LPS)-induced inflammation. Results: The extract contained eight peaks representing phenolic compounds and one peak representing riboflavin, with the corresponding mass spectrometry data documented. These biologically active compounds were purified by ultrafiltration using LC to determine their ability to target cyclooxygenase-2 (COX-2) and 2,2-diphenyl-1-picrylhydrazyl (DPPH). The results showed that significant compounds were identified, demonstrating a binding affinity for both COX-2 and DPPH. This suggests that the compounds showing excellent binding affinity for COX-2 and DPPH may be the main active ingredients. Vital inflammatory cytokines, including COX-2, inducible nitric oxide synthase (iNOS), mitogen-activated protein kinase (MAPK), and nuclear factor kappa B (NF-κB), were found to be down-regulated during the treatment. In addition, we revealed that the selected drugs exhibited potent binding capacity to inflammatory factors through molecular docking studies. In addition, we confirmed the presence of phenolic components in GLE extract and verified their possible anti-inflammatory and antioxidant properties. Conclusions: This study provided evidence for an efficient strategy to identify critical active ingredients from various medicinal plants. These data may serve as a baseline for further investigations of applying GLE in the pharmaceutical industry.

## 1. Introduction

*Glehnia littoralis* is a perennial herb found in sandy coastal regions. It is classified within the Apiales order, Apiaceae family, Apioideae subfamily, and Selineae tribe. This herb, discovered in regions such as Korea, Japan, and Taiwan, exhibits efficacy in the treatment of chronic bronchitis and alleviation of associated symptoms [1]. Research findings indicate that *Glehnia littoralis* extract has a preventive effect on memory disorders and neuroinflammation [2]. Another study found that *Glehnia littoralis* effectively treats acute and chronic skin inflammation [3]. Previous research has identified multiple phenolic compounds in the roots of *Glehnia littoralis*, known for their antioxidant and immunomodulatory properties [4,5]. Natural materials are employed in the synthesis of antioxidants and anti-inflammatory medications as a substitute for synthetic pharmaceuticals [6].

Inflammation serves as a protective mechanism in the body by combating pathogens and facilitating wound healing. Innate immune cells such as neutrophils, mast cells, fibroblasts, and macrophages become activated in response to infection. Activated macrophages release pro-inflammatory cytokines that exacerbate various inflammatory conditions [7]. Oxidative stress is intricately connected with the inflammatory response. The accumulation of reactive oxygen species leads to tissue and cellular injury, resulting in prolonged inflammation [8]. Reactive oxygen species trigger the activation of NF-κB by phosphorylating IκBα, resulting in heightened inflammation due to the transcription of inducible nitric oxide (NO) synthase [9]. The inflammatory response is the immune system’s reaction to a specific stimulus. The cell membrane of Gram-negative bacteria contains LPSs, which induce inflammatory responses in macrophages [10]. LPSs induce the release of the following inflammatory mediators: NO and prostaglandin E2 (PGE2) [11]. The levels of inflammatory cytokines, such as COX-2 and iNOS, are elevated upon activation of specific pathways [12]. This has prompted extensive research on inflammation related to lipopolysaccharides. Research indicates a potential connection between inflammation and various diseases, such as cancer, mental disorders, and heart conditions. In addition, addressing this persistent inflammation poses a growing challenge [13].

Synthetic drugs and immunosuppressants have the capability to modulate inflammation and immune responses; however, they have been associated with adverse effects such as abdominal discomfort, alopecia, emesis, and gastrointestinal disturbances [14]. The creation of novel anti-inflammatory medications from natural sources is becoming increasingly important. Studies have validated the antioxidant and anti-inflammatory characteristics of *Glehnia littoralis* [2,15]. However, the particular phenolic compounds accountable for the anti-inflammatory and antioxidant properties, along with their structural and binding characteristics for anti-inflammatory effects, remain unidentified.

The DPPH radical-based HPLC post-column assay is utilized for the evaluation and identification of natural antioxidants in botanical remedies [16]. In methanol, DPPH exhibits a significant absorption peak at 517 nm attributed to its unpaired electron. Antioxidants induce a change in color from purple to yellow by decreasing absorbance and creating a stable DPPH molecule through the donation of hydrogen to free radicals. The colorimetric assessment of DPPH is an uncomplicated technique for the post-column detection of radical-scavenging compounds [10]. The DPPH assay demonstrates ease of use, provides a consistent baseline, requires no expensive or unpredictable reagents, and enables the evaluation of both the radical-scavenging capability of a compound and its impact on the overall antioxidant capacity of a blend [17]. Thus, it seems that HPLC-DPPH is a reliable and effective method for quickly evaluating crude extracts that include antioxidants.

Various methods are utilized to identify chemical compounds present in *Glehnia littoralis*. These methods have not been utilized extensively across various samples; however, they have demonstrated limited sensitivity, inadequate reproducibility, challenges with self-quantification determination, and false positive outcomes. Some are still undergoing refinement. The preferred method for identifying compounds in various substances is ultra-filtration liquid chromatography–tandem mass spectrometry (UFLC-MS/MS) because of its rapid analysis and superior sensitivity and selectivity [18]. Therefore, it appears that UFLC-MS/MS is a viable method for identifying chemicals in *Glehnia littoralis* extract.

Molecular docking is used to predict the geometries of protein–ligand binding sites, which are frequently utilized in drug discovery and the prediction of functional sites on protein surfaces [19]. Molecular docking can be conducted by evaluating surface area binding and calculating the mutual energy interaction between the ligand and protein [20,21]. The measurement of binding affinity is distinct from the assessment of docking scores. It is crucial to confirm the results of molecular docking visually for validation purposes [22].

The objective of this research was to determine the phenolic compounds present in *Glehnia littoralis* leaves and investigate their influence on antioxidant and anti-inflammatory properties. Phenolic compounds within GLE were characterized utilizing HPLC-MS/MS, and their antioxidant properties were demonstrated utilizing DPPH and LC. The inhibition of inflammation via binding with COX-2 was verified utilizing HPLC-MS/MS. Subsequently, the anti-inflammatory activity of GLE was assessed in RAW264.7 cells displaying LPS-induced inflammation. This method may be time-consuming for researchers; however, it has the potential to pinpoint efficacious compounds. Natural products consist of active components, thereby rendering our approach valuable in the initial stages of drug discovery.

## 2. Material and Method

### 2.1. Plant Materials

*Glehnia littoralis* sourced from Yeongdeok, Gyeongsangbuk-do, on the East Coast of Korea, was provided for the experiment by the Animal Bio Resources Bank, a Nationally Designated Research Materials Bank (https://abrb.or.kr/index.php?PHPSESSID=2ff181b6f014eac4765a1642b188340d, accessed on 24 February 2023, QR code: 10210A). After quickly washing the leaves of the plant with water, we sliced it up and let it dry for 72 h at 56 °C in a dry oven. After that, it was kept at −20 °C in sealed polyethylene bags with silica gel when needed. Because of the nature of *Glehnia littoralis*, the stem and root are small in quantity, making it difficult to use, and only the leaves are used as the material. Therefore, in this paper, the leaves were used for extraction.

### 2.2. Reagents, Chemicals, and Standards

The DPPH reagent (CAS no. 1898-66-4), standard chemicals, and COX-2 enzymes were purchased from Sigma-Aldrich Corp. (St. Louis, MO, USA). A Millipore Co., Ltd. (Burlington, MA, N. Eng., USA) 30 kDa centrifugal ultrafiltration filter (YM-30) was purchased. Analytical-grade solvents and other compounds were all used (Duksan Pure Chemical Co., Ltd., Ansan, Gyeonggi-do, Republic of Korea). Gibco (BRL Life Technologies, Grand Island, NY, USA) supplied fetal bovine serum (FBS), Dulbecco’s modified Eagle’s medium (DMEM), phosphate-buffered saline (PBS), and antibiotics penicillin/streptomycin (P/S). Cell Signaling Technology (Danvers, MA, USA) provided the iNOS (cat. no. 13120S), COX-2 (cat. no. 12282S), JNK (cat. no. 9258S), *p*-JNK (cat. no. 4671S), ERK (cat. no. 4695S), *p*-ERK (cat. no. 4370S), p38 (cat. no. 8690S), *p*-p38 (cat. no. 9216S), p65 (cat. no. 8242S), *p*-p65 (cat. no. 3033S), IкBα (cat. no. 4812S), *p*-IкBα (cat. no. 2859S), and β-actin (cat. no. 3700S) antibodies. The secondary antibodies conjugated with horseradish peroxidase (HRP) against rabbit (cat. no. A120-101P) and mouse (cat. no. A90-116P) were supplied by Bethyl Laboratories, Inc. (Montgomery, AL, USA).

### 2.3. GLE Extraction Procedure and Phenol Component Purification

GLE was employed in a modified procedure to extract phenolic from plants [23]. For 4 days, 200 g of *Glehnia littoralis* leaves were extracted using 4 L of 70% methanol. The separation of the mixture was achieved by employing filter paper labeled as Whatman Qualitative No. 6. The mixture was condensed to 500 mL using a rotary evaporator (N-1110, Eyela, Tokyo, Japan) spinning at 100 rpm at 45 °C and reduced pressure. Subsequently, the concentrate was washed three times with 500 mL of hexane to eliminate any fatty particles. Three extractions were conducted using the residue remaining in the filtrate and 250 mL of ethyl acetate. At first, MgSO_4_ was utilized to desiccate remains to eliminate the highly polar components. Next, ethyl acetate and a silica gel solvent (40 cm, 2.5 cm) were used to elute the remains. In order to produce a mixed phenol powder, the solvent was evaporated under reduced pressure and then stored at −70 °C (8.8 g, or 5.5% of the dry raw *Glehnia littoralis* leaves).

### 2.4. HPLC and LC-MS/MS

The extract powder was mixed with 70% ethanol in a ratio of 1:100 to make a solution with a concentration of 1000 μg/mL for identifying the components. LC-MS/MS was carried out using an Ultra Quadrupole and a 1260 series HPLC system (Agilent Technologies, Inc., Santa Clara, CA, USA). The Time-of-Flight LC/MS/MS system (X500R) was operated in positive ion mode with the spray voltage set to −4.5 kV. A gradient system running at 0.5 mL/min was employed to analyze a Prontosil C18 column (length: 250 mm, inner diameter: 4.6 mm, particle size: 5 μm) (Bischoff Chromatography, Phenomenex Co., Ltd., Torrance, CA, USA). Acetonitrile and distilled water (DW) with 0.1% formic acid (solvent A, B) were utilized as solvents. The solvent B mobile phase conditions were as follows: 0–10 min at 10–15%, 20–30 min at 20–40%, 40–50 min at 70%, and 50–60 min at 95%. The analysis was conducted at 35 °C and a wavelength of 284 nm.

### 2.5. DPPH Binding HPLC Analysis to the Quantify Phenolic Compounds’ Primary Antioxidant Activities

Following the combination of GLE (2000 μg/mL) and DPPH (0.2 mg/mL) reagents in a 1:1 (*v*:*v*) ratio, the resulting mixture underwent a 15 min reaction period at ambient temperature [10]. The solution underwent filtration through a 0.45 μm filter prior to HPLC testing. Methanol was employed as a substitute for the DPPH reagent in the control group. The composition of DPPH-reactive compounds was determined by comparing the chromatographic peak values of DPPH-reacted samples with controls. This facilitated the identification of the primary antioxidant constituents within the GLE compounds.

### 2.6. UF-HPLC-Based Determination of the Reaction Between Phenol Compounds and COX-2

The anti-inflammatory potential of compounds was evaluated by their binding to COX-2 using a modified technique [24]. First, 20 μL of COX-2 (2U) was reacted with 100 μL of extract diluted to 2000 μg/mL (with 70% ethanol) in a water bath at 37 °C for 30 min. In the control group, COX-2 was inactivated using boiling water, whereas, in the experimental group, non-inactivated COX-2 was used in the reaction. The mixture was spun in a centrifuge with a 30 kDa cut-off ultrafiltration membrane (YM-30) at 10,000 rpm for 10 min at room temperature. Following centrifugation, unbound chemicals were eliminated by washing the solution 3 times with 200 μL of NE buffer (pH 7.9, 25 °C) (NEB (New England Biolabs), Ipswich, MA, USA) that did not pass through the filter. Following the washing process, the solution remaining on the surface exhibited a chemical that selectively binds with COX-2 and was unable to permeate the filter, as opposed to the filtered solution that contained a compound lacking affinity for COX-2 binding. Compounds that were not bound to COX-2 in the upper layer were subsequently dissolved in 80% acetonitrile for a duration of 10 min and subsequently separated by centrifugation at 10,000 rpm for 10 minutes (repeated 3 times). Subsequently, high-performance liquid chromatography (HPLC) was employed to analyze the filtrate underneath.

### 2.7. Measurement of Anti-Inflammatory Effects

#### 2.7.1. Cell Culture and Viability Assay

The RAW264.7 macrophage cells were provided by the American Type Culture Collection (ATCC) in Manassas, VA, USA. The cells were cultured in complete DMEM with 10% FBS and added with 100 U/mL penicillin and 100 μg/mL streptomycin (P/S). The cells were cultured at a temperature of 37 °C in a controlled environment with 5% CO_2_. RAW264.7 cells were seeded at a concentration of 1 × 10^4^ cells per well in 96-well plates and incubated for 12 h Following this, the cells were exposed to varying concentrations of GLE (0, 10, 25, 50, 75, 100, 250, 500, 750, and 1000 ng/mL) with or without 1 μg/mL of lipopolysaccharide (LPS) (Sigma-Aldrich, Merck KGaA, Burlington, VT, USA). After introducing 90 μL of media and 10 μL of MTT solution (5 mg/mL) into each well, the cells were incubated at 37 °C for four hours. The insoluble formazan crystals were dissolved with the use of DMSO. Finally, each sample underwent three consecutive runs, and the absorbance measurement of each well at 450 nm was obtained using a microplate reader (BioTek, Winooski, VT, USA) [7]. At concentrations of 25 and 50 ng/mL, no toxicity was observed; therefore, these concentrations were utilized in subsequent studies.

#### 2.7.2. Western Blot Analysis

After being seeded into 60 mm plates at a density of 1 × 10^6^ cells per well, RAW264.7 cells were treated with or without LPS 1 μg/mL (Sigma-Aldrich, Merck KGaA, Burlington, MA, USA) for 24 h at 37 °C, using 25 and 50 ng/mL of GLE. The treated cells were then lysed using radioimmunoprecipitation assay (RIPA) buffer (iNtRON Biotechnology, Seongnam, Gyeonggi, Republic of Korea), which includes a phosphatase inhibitor (Thermo Fisher Scientific, Waltham, MA, USA) and a protease inhibitor cocktail. The protein content of each cell lysate sample was measured using the bicinchoninic acid (BCA) test (Thermo Fisher Scientific, Waltham, MA, USA) in compliance with the manufacturer’s instructions. Equal amounts of protein (10 μg) were isolated using sodium dodecyl sulfate–polyacrylamide gel electrophoresis (SDS-PAGE) at a concentration of 10–15%. Following the creation of polyacrylamide gels, they were transferred to polyvinylidene fluoride (PVDF) membranes (ATTO Co., Ltd., Tokyo, Japan) using a semi-dry conveying system (JP/WSE-4040 HorizeBLOT 4M-R WSE-4045, Atto Corp., Tokyo, Japan). Afterward, the EzBlockChemi (ATTO Blotting System, Tokyo, Japan) was used to block the membranes for 2 h at ambient temperature. The membranes were then incubated with a 1:1000 diluted primary antibody overnight at 4 °C. The membranes were treated with 1:5000 diluted antirabbit and antimouse (cat. no. A120-101P, Bethyl Laboratory, Inc.) for 3 h at room temperature after being washed five times for 15 min with Tween 20 (TBS-T, pH 7.4). After that, the membranes were rewashed ten times for a total of two hours using TBS-T. ChemiDoc imaging equipment (Version 6.0) from Bio-Rad Laboratories, Inc. (Hercules, CA, USA), was used to take the pictures. The images were processed using the Bio-Rad application Image Lab 4.1. The detection method used Bio-Rad, Hercules, CA, USA’s enhanced chemiluminescence (ECL) buffer. The loading control was β-actin protein, and the Western blot pictures were quantified using the Image J program (U.S. National Institutes of Health, Bethesda, MD, USA) [7].

### 2.8. Molecular Docking Analysis

To execute the molecular docking analysis, the protein structure was obtained from PDB (https://www.rcsb.org/, accessed on 25 October 2023). Using the search ID 4Q3J (NF-кB), the 3D compound structures of chlorogenic acid (CID: 1794427), quercetin (CID: 5280343), and CPUY192018 (CID: 73330369) were reclaimed from PubChem (https://pubchem.ncbi.nlm.nih.gov/, accessed on 19 October 2023). AutoDock Vina and UCSF Chimera were used with default settings for docking analysis. The binding affinities were computed using total intermolecular energy and approximated free energy binding.

### 2.9. Statistical Analysis

The data are indicated as mean ± SEM. GraphPad Prism (version 8.0.1; GraphPad Software, Inc., La Jolla, CA, USA) was utilized for data analysis. This study employed one-way factorial analysis of variance (ANOVA) to ascertain the existence of momentous differences among the groups. After that, Dunnett’s multiple comparison tests were performed, and a statistically significant result was defined as *p* < 0.05. In comparison with the untreated, positive control group, ^#^ *p* < 0.05, ^##^
*p* < 0.01, and ^###^ *p* < 0.001, and * *p* < 0.05, ** *p* < 0.01, and *** *p* < 0.001 compared with the LPS-treated negative control group.

## 3. Results

### 3.1. Separation and Characterization of Phenols in GLE

The chemical compounds revealed in GLE were subjected to both quantitative and qualitative analysis using HPLC-MS/MS. HPLC analysis using GLE was repeated three times. The HPLC retention periods and UV-Vis spectra yielded nine peaks (Figure 1). At a wavelength of 248 nm, the peaks of eight phenolic compounds and riboflavin [25] were identified using HPLC chromatography. The eight phenol compounds were chlorogenic acid [26], rutin [27], Hyperin [28], Kaempferol-3-O-rutinoside [29], Astragalin [30], quercetin [31], cinnamic acid [32], and luteolin [33]. Fragmentation patterns served as the basis for the results. Appendix A presents the mass spectrometry data from published sources that were used to quantify the eight phenolic substances and riboflavin. Appendix A displays the results of the compound’s fragmentation prediction based on these findings.

### 3.2. Analyzing Antioxidant Phenolic Compounds for Opportunity in GLE

DPPH and HPLC were used in conjunction to identify potential antioxidant activity candidates among the phenolic components in the *Glehnia littoralis* extract. Prior to and during the DPPH procedure, the phenol components were compared and screened using the HPLC peak area values. As shown in Figure 1, once DPPH bound to GLE and the reaction occurred, the HPLC-MS/MS results showed that GLE included a variety of different bioactive chemicals. Table 1 illustrates the antioxidant impact of the phenol component that reacted vyingly with DPPH by showing a decrease in peak area after the reaction. The difference in area values (%) reflects the phenol component’s remarkable ability to scavenge radicals. Table 1 shows that quercetin and riboflavin had the highest relative peak area ratio differences, at 17.06% and 15.54%, respectively, followed by chlorogenic acid at 9.63%. Kaempferol-3-O-rutinoside and rutin showed the smallest difference in the change rate of binding peak area with DPPH at 2.63% and 1.14%.

### 3.3. Analyzing Anti-Inflammatory Phenolic Compounds for Prospective Use in GLE

The degree to which the eight specific phenolic compounds from *Glehnia littoralis* extract bind to COX-2, an inflammatory factor that explains the anti-inflammatory properties of each compound, is displayed in Figure 1 as peaks before and after the COX-2 reaction with UF-HPLC-based determination. The binding of phenol compounds from *Glehnia littoralis* extract to COX-2 resulted in an increase in the area value of the COX-2-activated peak relative to the COX-2-inactivated peak. Table 2 displays high relative area difference reaction rates for quercetin, chlorogenic acid, riboflavin, and cinnamic acid at 18.41%, 13.90%, 11.88%, and 11.21%. Kaempferol-3-O-rutinoside and rutin showed the smallest difference in the change rate of COX-2 binding peak area at 5.44% and 3.64%.

### 3.4. Effects of GLE on RAW264.7 Cell Viability

To evaluate the cytotoxicity of the extract, RAW264.7 was subjected to the 3-(3,4-dimethyl-thiazolyl-2)-2,5-diphenyl tetrazolium bromide (MTT) assay (Figure 2A,B). RAW264.7 cells were treated with GLE for 24 h at 0, 10, 25, 50, 75, 100, 125, 250, 500, 750, and 1000 ng/mL with or without 1 µg/mL LPS. Figure 3 shows that the extract is non-toxic at values of 25 and 50 ng/mL. Ultimately, concentrations of 25 and 50 ng/mL did not elicit any noticeable impact on RAW264.7 cells (Figure 2), thus justifying their selection for subsequent investigations.

### 3.5. Expression of COX-2 and iNOS in LPS-Induced RAW264.7 Cells

The inflammatory response involves COX-2 and iNOS proteins. The concentrations of GLE at 25 and 50 ng/mL showed a dose-dependent decrease in INOS and COX-2 (Figure 3). This suggests that GLE is effective in alleviating inflammatory responses.

### 3.6. Inhibition of NF-κB and MAPK Pathways in LPS-Induced RAW264.7 Cells

The NF-κB pathway affects multiple proteins in the immune system and has a significant function [34]. NF-κB contains *p*-p65 and *p*-IκBα, which fulfills that function. Furthermore, GLE tended to cause *p*-p65 to decline in a dose-dependent manner (Figure 4A). *p*-IκBα tended to decrease rapidly in response to 25 ng/mL of GLE (Figure 4B).

MAPK is crucial to immunological signaling. It includes *p*-JNK, *p*-ERK, and *p*-p38, which are essential to this signaling [35]. Their levels decreased following treatment with GLE at concentrations of 25 and 50 ng/mL (Figure 4C–E).

### 3.7. Molecular Docking of Quercetin, Chlorogenic Acid, Riboflavin, and CPUY192018 with NF-κB

Using UCSF Chimera software (version 1.16), ligand–protein docking was examined [36]. Figure 5A demonstrates that quercetin and NF-кB are present in the active site. Moreover, it was shown that many active sites promote ligand binding. It was found that NF-кB binds to quercetin through active sites ARG237, LEU236, CYS149, TYR227, and GLU184 (Table 3). The molecular binding energy score was found to be −7.5 kcal/mol.

Figure 5B depicts the occupation of both chlorogenic acid and NF-кB at the active site. Several active sites have been demonstrated to aid in ligand binding. The active sites involved in NF-кB’s binding to luteolin were identified as ASN240, ARG237, CYS149, GLY180, LEU236, and GLU184, as shown in Table 3. In the docking results, chlorogenic acid showed a molecular binding energy score of −6.8 kcal/Mol, indicating a lower interaction strength at the binding site compared with riboflavin.

Figure 5C demonstrates that CPUY192018, a well-known strong inhibitor, binds to the NF-кB active site as well. HIS183, CYS149, GLU184, LEU236, PRO147, ILE248, PHE146, ARG237, TYR227, ARG232, and GLU233 are among the proteins with which it interacts (Table 3). The molecule docking energy score was −7.7 kcal/mol.

## 4. Discussion

*Glehnia littoralis* is a perennial herb that can be commonly observed in coastal sand dunes across East Asia [37]. The dried roots can be used both as a dietary ingredient and a therapeutic element in nutritious food preparations. This herb has been studied and shown to have hepatoprotective, immunomodulatory, antioxidant, antibacterial, antifungal, anti-inflammatory, and anticancer properties [1,38].

As the limitations of conventional synthetic drugs have been revealed, the development of drugs borrowed from natural products, including traditional medications, is gaining momentum. It is known that free radicals and inflammation-related mediators are closely related to the development of diseases, and efforts to find antioxidant and anti-inflammatory substances are continuing [39]. To contribute to these efforts, this research investigated the antioxidant and anti-inflammatory effects of *Glehnia littoralis*, a herbal medicine used for bronchitis [40].

This study utilized GLE’s electron donation to DPPH as a method to evaluate its antioxidant properties. The assessment of antioxidant capacity is frequently conducted using DPPH, a consistent free radical that can be discolored and decreased through electron donation [41]. DPPH reacts with OH groups to form DPPH-H. Upon reduction, DPPH generates a colorless hydrazine compound known as DPPH-H, exhibiting a yellow coloration. The antioxidant activity measurement is determined based on the absorbance value at a wavelength of 519 nm, which is not within the yellow spectrum [42]. In this study, the DPPH area change values of quercetin, riboflavin, and chlorogenic acid compounds were found to be 17.06%, 15.54%, and 9.63% (Table 1). This indicates that quercetin, riboflavin, and chlorogenic acid are bound to DPPH in relatively high proportions, demonstrating a high antioxidant potential in GLE. Studies have shown that chlorogenic acid not only has antioxidant properties but also affects immune responses [43]. Furthermore, quercetin has antioxidant properties and affects obesity-related cirrhosis and retinopathy [44]. Other studies have also shown that riboflavin plays a role in antioxidant activity and in protecting vital organs against fluorosis [45,46]. In this study, GLE showed significant binding areas for quercetin, riboflavin, chlorogenic acid, and DPPH, suggesting its potential antioxidant capacity compared to previous studies.

Interestingly, despite being recognized as a potent antioxidant compound [47], rutin has the lowest representation in Table 1 (1.14%). Rutin demonstrates significant antioxidant properties upon individual component analysis. Nevertheless, this study was confirmed by inducing a competitive reaction in GLE, an extract consisting of various components. Rutin only showed lower competitiveness compared with the other components present in GLE.

In this study, the anti-inflammatory activity was quantified using the COX-2 binding ability of GLE. It is known that inhibition of COX directly stops the production of prostaglandins (PGs), which play an important role in inflammation [48]. In addition, inhibition of COX-2, a key enzyme in the inflammatory process, shows anti-inflammatory and analgesic effects [49]. Therefore, it is important to examine the anti-inflammatory potential through the combination of COX-2 and GLE components. The relative area difference response rates were in the order of quercetin, chlorogenic acid, riboflavin, and cinnamic acid (18.41%, 13.90%, 11.88%, and 11.21%) (Table 2). According to the study, quercetin and quercetin metabolites obtained by consuming quercetin provide powerful antioxidant and anti-inflammatory effects [50]. Chlorogenic acid also attenuated other inflammation-related markers, such as inflammatory cytokines (including IL-1β and TNF-α) and IL-6, without cytotoxicity [51]. Additionally, a previous study demonstrated that cinnamic acid effectively inhibits the production of NO, TNF-α, and PGE2 [52]. Therefore, the results of this study demonstrate that quercetin, chlorogenic acid, riboflavin, and cinnamic acid have significant binding areas with COX-2, suggesting potential anti-inflammatory effects compared to previous studies.

Numerous investigations have shown that pro-inflammatory cytokines are released when LPS-induced inflammation activates the MAPK and NF-кB pathways [53]. Important pro-inflammatory cytokines, including COX-2 and iNOS, play a role in regulating inflammation during this phase [54]. Previous studies have shown that bioactive substances can act as effective anti-inflammatory agents in RAW264.7 cells exposed to LPS, inhibiting the pro-inflammatory cytokines COX-2 and iNOS [55]. Our objective was to investigate the impact of GLE on iNOS and COX-2. Our findings demonstrate that iNOS and COX-2 expression levels were reduced in a dose-dependent manner with GLE, specifically at concentrations of 25 and 50 ng/mL (Figure 3A,B). Inhibition of COX-2 and iNOS can mitigate excessive inflammation [56,57]. Additionally, the decrease in COX-2 and iNOS caused by GLE at doses as low as 25 and 50 ng/mL suggests that a drug may have a significant effect even at lower doses. Consequently, this decrease indicates that GLE can be expected to have anti-inflammatory properties by reducing proteins related to inflammation.

Complex signaling pathways like NF-κB/MAPK have been demonstrated to initiate inflammatory responses [58]. NF-κB controls the activation of inflammasomes and influences the production of pro-inflammatory genes [59]. The administration of GLE resulted in the downregulation of NF-κB transcription factor phosphorylation in a dose-dependent manner. These outcomes were also noted in the MAPK signaling cascade below. MAPKs are crucial for the activation of inflammatory cytokines and the release of inflammatory chemokines [60]. p38 is one of the three major MAPKs (JNK, ERK (1/2)), and it plays a role in controlling the production of inflammatory regulators [61]. Hence, a potential approach to treating disorders linked to inflammation involves decreasing protein expression connected to NF-κB/MAPK signaling. In the NF-κB/MAPK signaling pathway, administration of GLE was observed to decrease the phosphorylation of p38, ERK (1/2), and JNK, which were elevated in the presence of LPS stimulation (Figure 4). The potential efficacy of low doses (25 and 50 ng/mL) in reducing factors of the NF-κB and MAPK pathways, similar to COX-2 and iNOS, may be significant for inflammation treatment.

Upon comparison of Table 1 and Table 2, it is evident that the docking scores of quercetin and chlorogenic acid closely resemble that of CPUY192018. CPUY192018 is known to be a potent inhibitor that relieves kidney inflammation and is used as a positive control [62]. Riboflavin was excluded from the docking section because it is not a phenolic compound. Based on the results of DPPH and COX-2 binding HPLC, quercetin, and chlorogenic acid are candidate phenolic compounds that are expected to have large peak-area change ratios and exhibit antioxidant and anti-inflammatory properties (Table 1 and Table 2). This indicates that the two substances may have antioxidant and anti-inflammatory properties akin to those of the positive control group. Recall that determining the precise binding affinity is not the means by which the molecular docking procedure outcomes are evaluated. The primary goal in this case is to compare the outcome values between the structures; the absolute docking result value is not significant. Furthermore, the binding structure may not always bind even when a high value is obtained. Therefore, visual confirmation of structural binding is essential for validating the molecular docking result [63]. These data were analyzed to discover which phenol compounds in GLE are more strongly linked to proteins involved in inflammation prevention, resulting in docking scores. As shown in Table 3, when compared with CPUY192018, which is known to be a strong inhibitor, quercetin did not show a significant difference in the results, and chlorogenic acid showed a difference, but not a significant difference, indicating that GLE, which is composed of several plant extracts, has strong anti-inflammatory and antioxidant properties. This suggests that GLE may become an increasingly useful anti-inflammatory drug.

Based on the results, we obtained the docking scores for the binding of phenol compounds of GLE, which have a significant effect on anti-inflammatory proteins. These DPPH, COX-2 binding, and docking results suggest that GLE containing various extracts has strong antioxidant and anti-inflammatory effects. In addition to the anti-inflammatory research results of *Glehnia littoralis* shown in several other studies [3,6,64], our in vitro findings and anticipated docking data point to GLE as a conceivable anti-inflammatory medication.

## 5. Conclusions

In this study, GLE is defined as an extract obtained from the leaves of *Glehnia littoralis*. Phenolic compounds were identified in the GLE extract, demonstrating antioxidant and anti-inflammatory properties. Although each compound displayed antioxidant and anti-inflammatory properties, it is possible to anticipate the synergistic effects of multiple phenolic compounds.

This study analyzed the phenolic compounds found in GLE and identified those with noticeable antioxidant and anti-inflammatory properties through DPPH, COX-2, and HPLC-based binding analysis. Furthermore, examination of the compounds with NF-κB through molecular docking analysis revealed that the components of GLE showed high binding scores in terms of structural affinity (Figure 6).

Therefore, the presence of phenolic compounds, antioxidant properties, anti-inflammatory effects, and molecular structural associations in these extracts suggest that GLE holds promise as a therapeutic agent for inflammation-related pathways because of its structural affinity and antioxidant effects.

## Figures and Tables

**Figure 1 nutrients-16-03656-f001:**
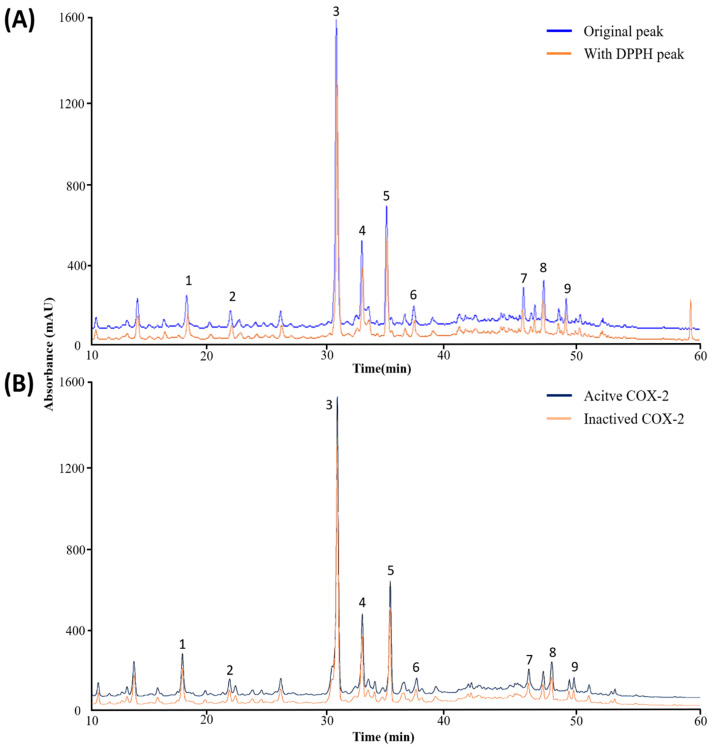
The phenolic compounds found in the HPLC chromatograms of GLE. The chromatogram following the reaction with DPPH solution is shown by the orange line in (**A**), and the chromatogram following the reaction with COX-2 is shown by the blue line in (**B**). In (**B**), active COX-2 is generated through the reaction of GLE with activated COX-2, while inactive COX-2 is produced through the reaction of GLE with inactive COX-2. The first chromatogram at the beginning of GLE is represented by the blue line. The detected compounds at the 284 nm wavelength are chlorogenic acid (1), riboflavin (2), rutin (3), Hyperin (4), Kaempferol-3-O-rutinoside (5), Astragalin (6), quercetin (7), cinnamic acid (8), and luteolin (9).

**Figure 2 nutrients-16-03656-f002:**
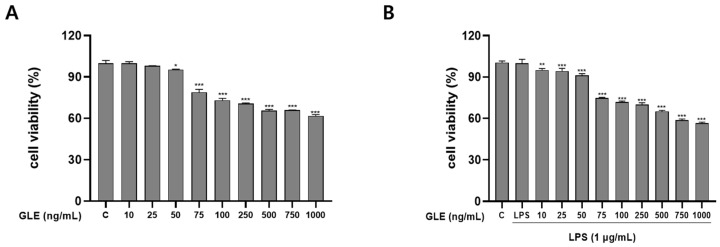
GLE’s cytotoxic effects on RAW264.7 cells. RAW 264.7 cells underwent an hour of pretreatment at 37 °C with or without LPS (1 μg/mL). GLE (0, 10, 25, 50, 75, 100, 250, 500, 750, and 1000 ng/mL) was then applied to the cells over a 24 h period at 37 °C. C is treated with only the medium, without GLE and LPS treatment, as control cells in RAW 264.7 cells. (**A**) GLE’s cytotoxic effect on RAW264.7 is not produced by LPS. * *p* < 0.05, *** *p* < 0.001 vs. the control group. (**B**) Cytotoxic effect of GLE on LPS-induced cell viability in RAW264.7 cells. ** *p* < 0.01, *** *p* < 0.001 vs. the LPS-treated group. RAW264.7 cells were treated with GLE (0, 25, and 50 ng/mL) at determined concentrations for 24 h.

**Figure 3 nutrients-16-03656-f003:**
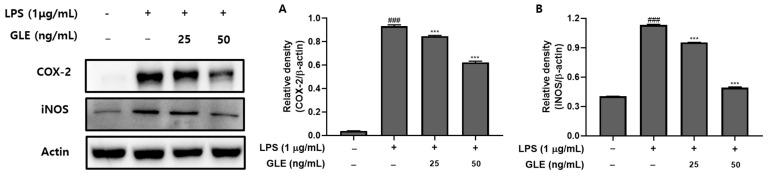
GLE-induced suppression of inflammatory factors in RAW264.7 cells caused by LPS. (**A**) The related density of COX-2 and (**B**) iNOS. The standard error of the mean (SEM) and mean for each of the three trials are displayed in relation to the control group. ^###^
*p* < 0.001 vs. the untreated group; *** *p* < 0.001 vs. the LPS-treated group.

**Figure 4 nutrients-16-03656-f004:**
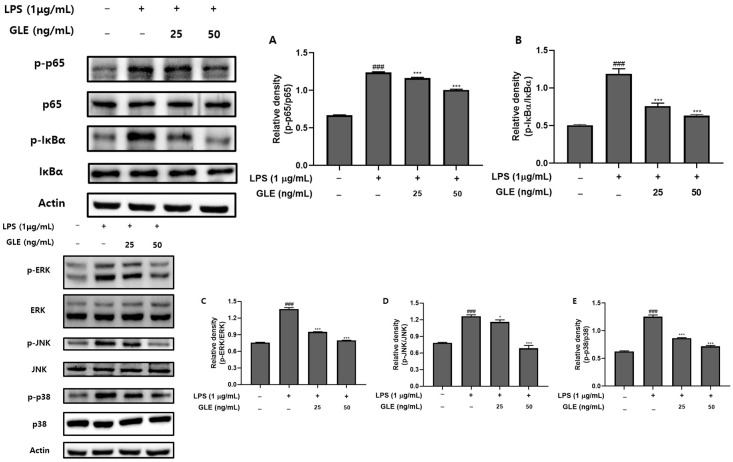
GLE-induced suppression of the NF-κB and MAPK pathways in LPS-induced inflammatory RAW264.7 cells. GLE (0, 25, and 50 ng/mL) was added to the RAW264.7 cells at the specified doses, and the treatment lasted for 24 h. (**A**) *p*-p65 relative density, (**B**) *p*-IκBα relative density, (**C**) *P*-ERK relative density, (**D**) *p*-JNK relative density and (**E**) *p*-p38 relative density. The results of three independent experiments are presented as mean and standard error of the mean (SEM), relative to the control group. ^###^
*p* < 0.001 vs. the untreated group, * *p* < 0.05, *** *p* < 0.001 vs. the LPS-treated group.

**Figure 5 nutrients-16-03656-f005:**
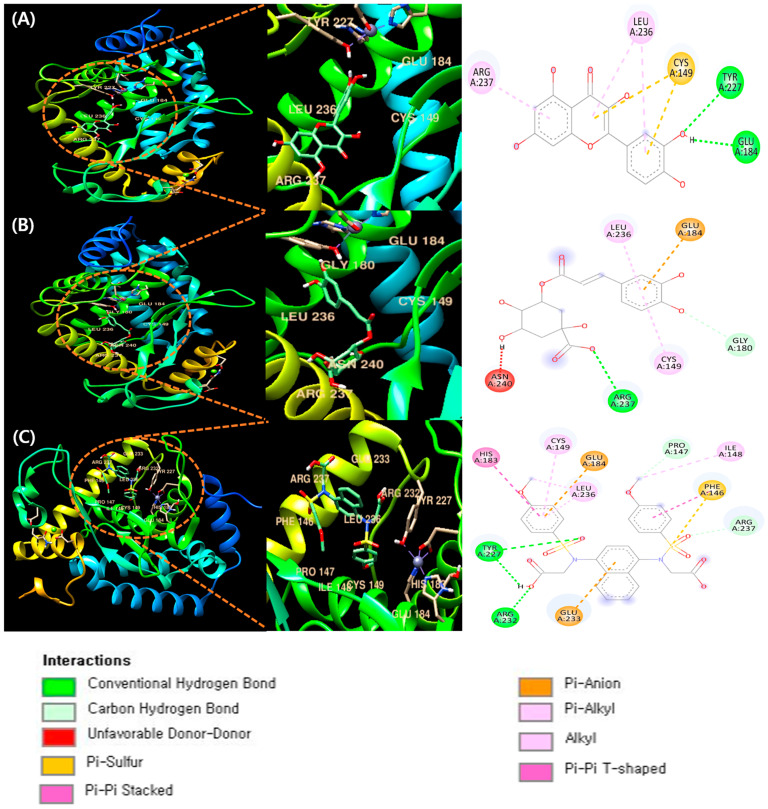
Molecular docking study of phenolic chemicals and NF-κB in GLE. The 3D structure of NF-κB bound efficiently to (**A**) quercetin, (**B**) chlorogenic acid, and (**C**) CPUY192018. Descriptions of peptide interactions are indicated by color and type of interaction below the docking diagram.

**Figure 6 nutrients-16-03656-f006:**
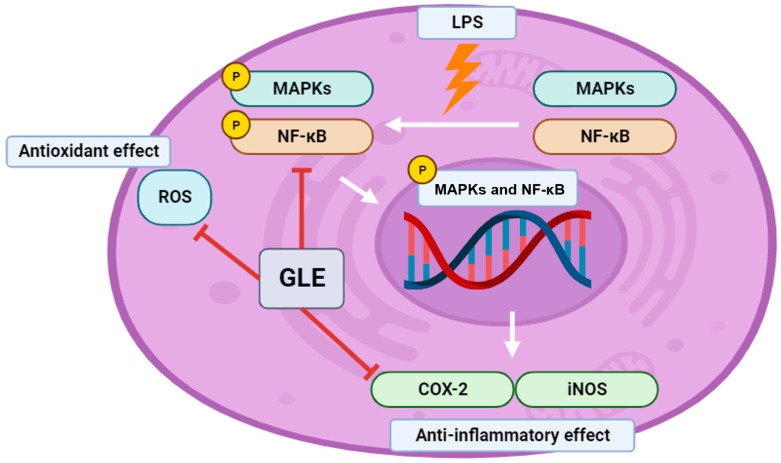
Schematic illustration of anti-inflammatory effects and antioxidant effects of GLE in RAW264.7 cells. P is phosphorylation. This diagram was created by BioRender.

**Table 1 nutrients-16-03656-t001:** Antioxidant capacity of the screened GLE compounds.

Peak No.	Compound	Initial Area	Area after DPPH Reaction	Reactive Area /(%)
1	Chlorogenic acid	2934.80 ± 8.95 ^F^	2652.07 ± 24.10 ^F^	282.73 ± 15.23 ^BC^ (9.63 ± 0.55 ^D^)
2	Riboflavin	1522.30 ± 4.37 ^B^	1285.74 ± 14.76 ^B^	236.56 ± 18.15 ^B^ (15.54 ± 1.16 ^E^)
3	Rutin	24,366.93 ± 25.31 ^I^	24,088.97 ± 37.21 ^I^	277.97 ± 15.24 ^BC^ (1.14 ± 0.06 ^A^)
4	Hyperin	6031.30 ± 33.05 ^G^	5712.57 ± 27.86 ^G^	318.73 ± 7.66 ^C^ (5.28 ± 0.11 ^BC^)
5	Kaempferol-3-O-rutinoside	7983.77 ± 14.59 ^H^	7773.50 ± 19.12 ^H^	210.27 ± 4.82 ^B^ (2.63 ± 0.06 ^AB^)
6	Astragalin	1811.87 ± 18.54 ^C^	1750.63 ± 39.93 ^D^	61.23 ± 21.39 ^A^ (3.39 ± 1.21 ^AB^)
7	Quercetin	1916.20 ± 33.11 ^D^	1589.43 ± 44.01 ^C^	326.77 ± 13.37 ^C^ (17.06 ± 0.94 ^E^)
8	Cinnamic acid	2597.03 ± 41.86 ^E^	2386.20 ± 79.36 ^E^	210.83 ± 111.40 ^B^ (8.08 ± 4.14 ^CD^)
9	Luteolin	1144.63 ± 26.29 ^A^	1061.77 ± 36.10 ^A^	82.87 ± 15.12 ^A^ (7.25 ± 1.41 ^CD^)

All values are mean ± SD (n = 3). ^A–I^ Means with different superscripts in the same column are significantly different at *p* < 0.05 based on Duncan’s multiple range tests.

**Table 2 nutrients-16-03656-t002:** The anti-inflammation potential of the screened GLE compounds.

Peak No.	Compound	With Active COX-2 Area	With Inactive COX-2 Area	Area Reacted with COX-2 /(%)
1	Chlorogenic acid	2323.42 ± 20.16 ^F^	2000.50 ± 17.43 ^F^	322.92 ± 3.11 ^E^ (13.90 ± 0.06 ^H^)
2	Riboflavin	860.20 ± 14.92 ^B^	763.71 ± 10.61 ^B^	96.49 ± 6.52 ^B^ (11.21 ± 0.61 ^F^)
3	Rutin	15,927.68 ± 81.61 ^I^	15,347.92 ± 52.82 ^I^	579.75 ± 43.02 ^F^ (3.64 ± 0.26 ^A^)
4	Hyperin	4302.05 ± 27.86 ^G^	4010.30 ± 23.53 ^G^	291.74 ± 5.82 ^D^ (6.78 ± 0.10 ^C^)
5	Kaempferol-3-O-rutinoside	6328.96 ± 35.06 ^H^	5984.61 ± 32.84 ^H^	344.35 ± 16.61 ^E^ (5.44 ± 0.25 ^B^)
6	Astragalin	945.17 ± 18.44 ^C^	862.35 ± 17.98 ^C^	82.81 ± 0.60 ^B^ (8.76 ± 0.13 ^D^)
7	Quercetin	1250.04 ± 26.12 ^D^	1019.96 ± 21.12 ^D^	230.08 ± 5.65 ^C^ (18.41 ± 0.19 ^I^)
8	Cinnamic acid	1866.91 ± 9.90 ^E^	1645.05 ± 14.03 ^E^	221.86 ± 4.13 ^C^ (11.88 ± 0.29 ^G^)
9	Luteolin	439.18 ± 5.80 ^A^	393.86 ± 6.98 ^A^	45.32 ± 1.18 ^A^ (10.32 ± 0.40 ^E^)

All values are mean ± SD (n = 3). ^A–I^ Means with different superscripts in the same column are significantly different at *p* < 0.05 based on Duncan’s multiple range tests.

**Table 3 nutrients-16-03656-t003:** Molecular docking studies of quercetin, chlorogenic acid, riboflavin, and CPUY192018 with the NF-кB complex and their binding energies.

Binding Ligand	Interacting Amino Acid Residue	Docking Score
Quercetin	ARG237, LEU236, CYS149, TYR227, GLU184	−7.5 kcal/mol
Chlorogenic acid	ASN240, ARG237, CYS149, GLY180, LEU236, GLU184	−6.8 kcal/mol
CPUY192018	HIS183, CYS149, GLU184, LEU236, PRO147, ILE248, PHE146, ARG237, TYR227, ARG232, GLU233	−7.7 kcal/mol

## Data Availability

The original contributions presented in the study are included in the article, further inquiries can be directed to the corresponding author.

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
