# Peer review of "Antioxidant and Anti-Inflammatory Properties of Conceivable Compounds from Glehnia littoralis Leaf Extract on RAW264.7 Cells"

_nutrients, 2024, doi:10.3390/nu16213656_

Round 1
Reviewer 1 Report
Comments and Suggestions for Authors
Article
Phenolic Compounds in Glehnia littoralis Extract (GLE) on RAW264.7 Cells: Possible Antioxidant and Anti-Inflammatory Properties
A brief summary
The research presented in the article is of interest, but the article itself requires a significant number of corrections, additions and clarifications. Furthermore, the text requires attention to ensure the language is correct in many aspects. The authors employ terminology that is not commonly utilized within this field. The descriptions are inadequate, bizarrely, and appear to have been poorly translated. It would appear that the authors lack prior experience with medicinal plants, as well as with the herbal raw materials they provide, their evaluation of these materials, and chromatography.
Broad comments
1. The paper promises to be of interest, presenting the composition and activity of an extract obtained from the leaves of Glehnia littoralis, an intriguing plant from the Apiaceae family. It is regrettable that the description of the method used to obtain the raw herbal material, the tests that were conducted, and the subsequent description of the results and conclusions were not presented in a manner that adheres to the standards of scientific rigor.
2. The selection of subject matter and the methodology employed in the assessment are to be commended. Nevertheless, the methodology for testing the oxidative capacity of the individual extract components by adding a DPPH solution prior to their separation on a chromatographic column remains unclear.
3. This work has the potential to be both interesting and valuable, but it would be beneficial to approach it with a renewed perspective.
4. The Methods section is written in a style that is difficult to comprehend, contains incorrect terminology and inaccurate descriptions of the research conducted. The text should be rewritten in its entirety from the beginning in order to ensure clarity and accuracy.
5. The conclusions are presented in a manner that is both clumsy and naïve.
6. The references appear to have been selected with care.
7. Additional comments and suggestions can be found below.
Specific comments
1. It seems that the abbreviation GLE in the title is unnecessary; it was adopted for the purpose of this article and is unlikely to be widely and commonly used. On the other hand, it would be beneficial to include the name of the solvent with which this extract was obtained, as this defines its possible composition (it contains chemical compounds that dissolve in it – eg. riboflavin is a water-soluble vitamin).
2. 'Keywords' is an excellent place to insert phrases about ongoing research that is not included in the title. E.g. 'COX-2', 'iNOS' and 'NF-кB' or ' instead of 'antioxidant', ' ' and 'Glehnia littoralis extract'.
The sentence in lines 45-47 is somewhat unclear.
Lines 94-95 . Is UFLC ultra-filtration liquid chromatography, or is it rather ultra-fast liquid chromatography? Ultrafiltration liquid chromatography, is more like UF-HPLC.
Lines 122-124 Who has confirmed the species identity? Is there a voucher specimen?
Lines 121-249. Please ensure that the text is coherent and linguistically accurate , and that the terminology used is appropriate.
Lines 252. In the context of plants and plant raw materials, the terms "compounds," "chemical compounds," and "active compounds" are preferred over "chemicals."
Comments on the Quality of English Language
The text of the review contains several instances where the use of language is discussed.
Author Response
Dear Reviewer 1
We appreciate your advice.
We have made revisions to the manuscript as per your advice.
Kindly review the attached file, revised manuscript, and supplementary file.
Kind regards
Min Yeong Park

Reviewer 2 Report
Comments and Suggestions for Authors
The manuscript entitled “Phenolic Compounds in Glehnia littoralis Extract (GLE) on RAW264.7 Cells: Possible Antioxidant and Anti-Inflammatory Properties” investigated the chemical composition of Glehnia littoralis extract, as well as the evaluation of antioxidant and anti-inflammatory effects.
The manuscript is quite interesting. However, major revisions should be made in order to be published in Nutrients journal, and the manuscript should be completed and/or modified taking into account the suggestions from attached file.

Minor editing of English language is required
Author Response
Dear Reviewer 2
We are grateful for your advice and comments.
We have made revisions to the manuscript as per your advice.
Please review the manuscript, supplementary file, and attached document.
Kind regards
Min Yeong Park

Round 2
Reviewer 1 Report
Comments and Suggestions for Authors
The corrections made by the authors, and the fact that they wrote most of the article from the beginning, have had a positive effect on its quality. It seems that it can now be published.
Personally, I will have to see how the approach of studying DPPH scavenging with BEFORE chromatographic separation of the individual extract components works in practice.